# Simulation of Multi-Phase Flow to Test the Effectiveness of the Casting Yard Aspiration System

**Serghii Lobov** [1,\*] , **Yevhen Pylypko** [2] , **Viktoriya Kruchyna** [1] **and Ihor Bereshko** [1]

[1] Department of Ecology and Technogenic Safety, National Aerospace University "Kharkiv Aviation Institute", 61070 Kharkiv, Ukraine; v.kruchyna@khai.edu (V.K.)

[2] Ukrainian Scientific and Technical Center of the Metallurgical Industry "Energostal", 61070 Kharkiv, Ukraine; pilipkoev@gmail.com

\* Correspondence: s.lobov@khai.edu

**Abstract:** The metallurgical industry is in second place among all other industries in terms of emissions into the atmosphere, and air pollution is the main cause of environmental problems arising from the activities of metallurgical enterprises. In some existing systems for localization, in the trapping and removal of dust emissions from tapholes and cast-iron gutters of foundries, air flow parameters may differ from the optimal ones for solving aspiration problems. The largest emissions are observed in the area of the taphole (40–60%) and from the ladles during their filling (35–50%). In this paper, it is proposed to consider a variant of the blast furnace aspiration system with the simultaneous supply of a dust–gas–air mixture from two-side smoke exhausters and two upper hoods with two simultaneously operating tapholes, that is, when the blast furnace operates in the maximum emissions mode. This article proposes an assessment of the effectiveness of the modernized blast furnace aspiration system using computer CFD modeling, where its main parameters are given. It is shown that the efficiency of dust collection in the proposed system is more than 90%, and the speed of the gas–dust mixture is no lower than 20 m/s, which prevents gravitational settling on the walls. The distribution fields of temperatures and velocities are obtained for further engineering analysis and the possible improvement of aspiration systems.

**Keywords:** aspiration system; blast furnace; casting yards; multi-phase flow processes; dust pollution





## 1. Introduction

It is air pollution that is the main cause of environmental problems arising from the activities of metallurgical enterprises. Emissions from pipes lead to soil pollution, destruction of vegetation and the formation of man-made wastelands around large factories. In addition, the environmental problems of domestic metallurgy are exacerbated due to high wear and tear of equipment and outdated technologies. In many countries, including Ukraine, about 70% of all capacities of the metallurgical industry are worn out, outdated and unprofitable, while metallurgy (ferrous and non-ferrous) accounts for about a third of all industrial emissions into the atmosphere [1–3].

Similar problems are observed in many countries where ferrous metallurgy is developed [4–13]. As a rule, carbon oxides (67.5%), suspended dust particles (15.5%), sulfur dioxide (10.8%) and other compounds are predominant in atmospheric emissions. The sum of the emissions of pollutants into the atmosphere can be 83.6% of the total volume of volatile pollutants. In Ukraine, for example, the population of large industrial cities and adjacent areas suffers primarily from the concentrated location of metallurgical industry enterprises. It is in these cities that there is a trend towards an increase in the incidence of environmentally dependent diseases (especially in children) [3–5].

When carrying out technological processes of casting iron in closed volumes, as is the case in various furnaces, the main part of dust and gas emissions is systematically

re-moved through gas exhaust paths and chimneys. In conditions when one or another technological process is open, an important place in the fight against air pollution is occupied by the prevention of dust and gas emissions by suppressing them in places of formation. Depending on the specific conditions of the foundry process, dust and gas emissions can be suppressed in various ways [6–8], but the most common is the method of capturing dust and gas emissions using aspiration systems with exhaust hoods, side exhausts, etc. To purify gases from chemical gaseous impurities, the following three methods can be used: absorption, adsorption and the transformation of gaseous impurities into a solid or liquid state [9].

The proposed task of computing the aspiration system was carried out as part of a project to improve the aspiration system of a blast furnace with an increase in its design capacity in terms of the amount of air being cleaned. The project was carried out for three years, including a technological audit of the blast furnace operation modes, a technological audit of the aspiration system, and a construction survey of gas duct structures. The parameters of the aspirated air before and after the treatment equipment were also measured, which included the following: productivity, velocity, dustiness, temperature, rarefaction in the gas duct, and chemical composition.

This information largely determined the initial data for calculating and modifying the geometry of the system.

Preparations for the reconstruction of the aspiration system began in 2022. At the moment, a part of the existing gas duct has been dismantled, exhaust hoods have been installed and a part of the designed branches of the gas duct has been laid in the places where they go through new air ducts. Unfortunately, at the moment it is not possible to continue work on the construction of the aspiration system.

The relevance of this work is due to the fact that, at the moment, at a metallurgical plant with the foundry yard under research, the existing aspiration system has a number of significant imperfections:

- Initially, the required amount of aspiration air was incorrectly calculated to capture the dust–gas–air mixture released at the points of release.
- The cross-sectional area of the gas ducts was chosen incorrectly, and therefore the velocity of the dust–gas–air mixture in some areas has a velocity of less than 20 m/s, while dust particles are accumulated in the gas ducts, and in some cases it is more than 35 m/s, and as a result there is abrasion of the structural elements [14–16].
- The main line of gas ducts was laid with a large number of turns, which are not curved along the radius, but at an angle of 90°, forming many stagnant zones in which dust accumulates.
- All existing gas ducts and local exhausters were laid below the marks at which the dust–gas–air mixture was released, which reduced the efficiency of the system as a whole, since the gases had a high temperature and raised sharply from the point of release.
- Local aspiration hoods have a small cross-sectional area, which causes emissions to escape past the hoods.
- The temperature regime of the system was incorrectly determined, which led to errors in the selection of structural materials. As a result, the gas duct walls were deformed and the connections lost their tightness.
- The existing system was designed only for the standard mode of operation—the discharge of cast iron on one taphole. In this case, the maximum release of the dust–gas–air mixture occurs sequentially—at the beginning of the process, when the taphole is opened, and then after some time when the taphole is blown. The non-standard mode of operation with simultaneous discharge of cast iron from two tapholes and parallel separation of the dust–gas–air mixture for the existing system was not considered during its development.

The proposed modernized aspiration system eliminates these imperfections. The scheme of the modernized aspiration system is shown in Figures 1 and 2. The total size of the foundry yard is $100 \times 100$ m.

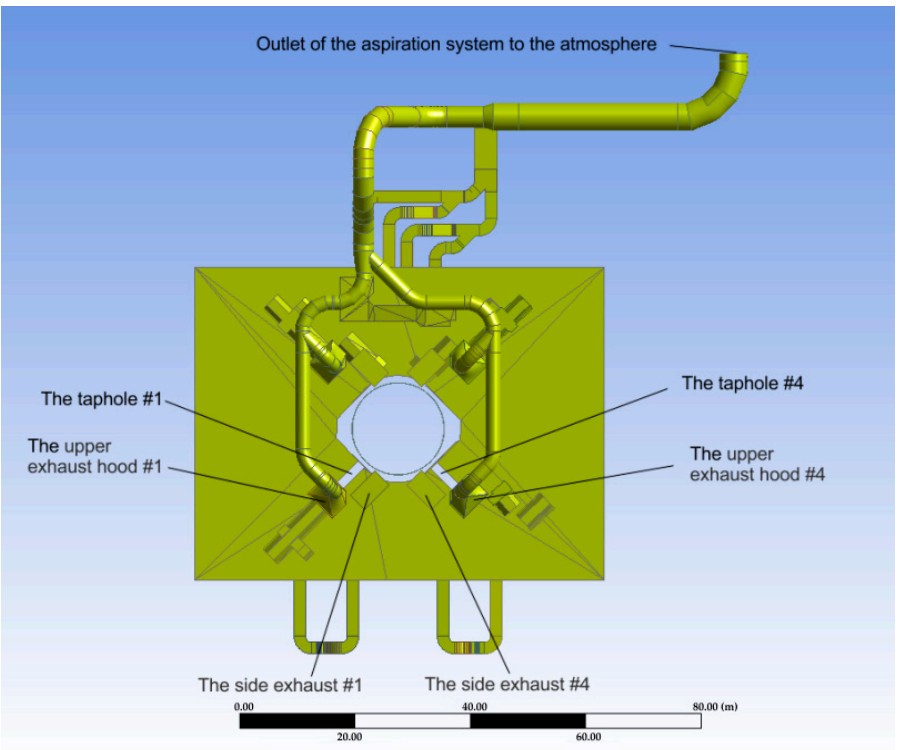

**Figure 1.** Scheme of the proposed aspiration system and part of the casting yard with places of emissions (top view).

It is obvious that the gas duct line also has a fairly large number of turns, but with smooth bends with a large bend radius, which eliminates the formation of stagnant zones. Local exhausters are located above the emission zones, directly along the trajectory of the dust–gas–air mixture. A new geometry has been developed and the cross-sectional area of aspiration hoods has been increased to completely remove the dust–gas–air mixture.

Mainly, the proposed system was designed for the simultaneous parallel operation of two tapholes (#1 and #4 on Figure 1). When computing, it was taken into account that on one taphole, cast iron is tapped, and on the other taphole, it is completed and the taphole is blown. At this moment, the maximum amount of the dust–gas–air mixture enters the aspiration.

Based on accumulated practical experience, specialists involved in the development of aspiration systems chose the most unfavorable operating mode with the simultaneous operation of two tapholes—tapholes #1 and #4. These entrances were chosen as the most distant from the aspiration unit (bag filter in the outlet of the system) and having the longest and geometrically heterogeneous route.

To check the geometry calculations and the effectiveness of the aspiration system, it was necessary to perform mathematical CFD modeling of the entire system. In other words, the main goals and tasks of modeling were as follows:

- Creation of a 3D mathematical model of the movement of a dust–gas–air mixture when ejected from two zones during the parallel operation of two tapholes when draining cast iron on one and blowing on the other.
- Checking the aerodynamic mode of operation of the aspiration system to identify zones in the gas duct with a reduced flow velocity and structural elements with a large coefficient of local resistance.

-　Determination of the concentration of coarse aerosols in the flow at the inlet to the gas duct and the gas duct itself.
-　Visual modeling of the aerodynamic field of suction plumes of local exhausters, calculated dimensions and geometry to determine their efficiency in terms of the amount of dust–gas–air mixture captured.
-　Creation of a flow map of temperature and velocity at the inlet to the gas duct and in the gas duct itself for the selection of structural materials.

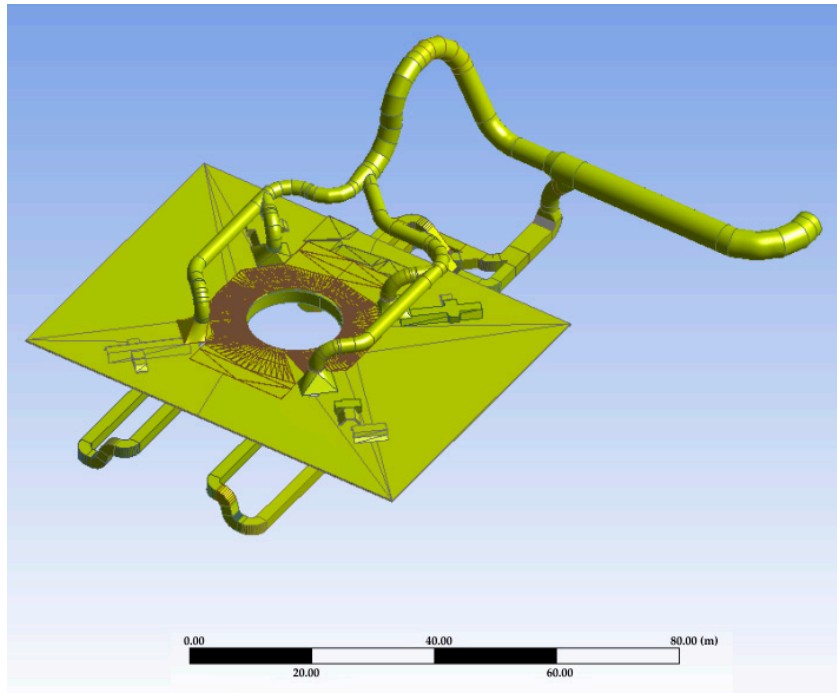

**Figure 2.** Scheme of the proposed aspiration system and part of the casting yard (isometric view).

## 2. Initial Data and Method of Modelling and Computing

The cast-iron taphole is a rectangular channel 250–300 mm wide and 450–500 mm high. The channel is made in chimney refractory masonry at a height of 600–1700 mm from the surface of the hearth. Channels for slag holes are laid out at a height of 2000–3600 mm. The channel of the cast-iron taphole is usually closed with a refractory mass. A cast-iron taphole is opened by drilling a hole with a diameter of 50–60 mm with a drilling machine. After the release of cast-iron and slag (on modern large blast furnaces, the release of cast iron and slag is carried out through cast-iron tapholes), the holes are clogged with an electric gun. The tip of the electric gun is inserted into the taphole and a taphole refractory mass is fed into it from the gun under pressure. The slag taphole at the blast furnace is protected by water-cooled elements, which are collectively called slag stoppers and a lever design with a remotely controlled pneumatic actuator.

Large volume blast furnaces (3200–5500 m$^3$) are equipped with four cast-iron tapholes operating alternately and one slag taphole (Figures 1 and 3).

When performing technological operations for the production of cast iron, a significant number of mechanical particles (dust), various gases and thermal energy are released into the air. The gases escaping from the taphole and partly above the main gutter form a fairly powerful and stable thermal jet adhering to the furnace casing and rising upwards to the ceiling of the casting yard.

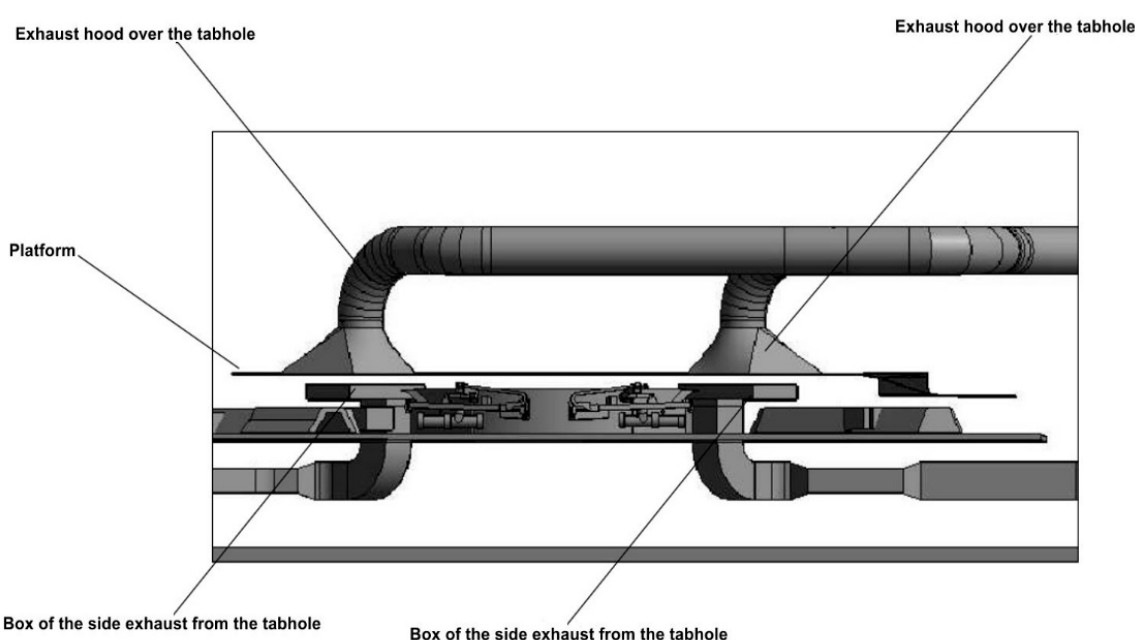

**Figure 3.** Aspiration system (side elevation) in casting yard.

Basic data for modeling are as follows:

1.  Productivity at the outlet of the aspiration system: 1,400,000 m$^3$/h.
2.  Dust content of aspirated air at the starting point: 5 . . . 6 g/m$^3$.
3.  The temperature of the dust and gas flow at the starting zone is 350 °C.
4.  Ambient temperature: +30 °C.
5.  Air temperature of the working area: +30 °C.
6.  Figure 4 shows the chemical composition of dust emitted from the taphole by weight. Since dust contains (by mass) most of all Fe$_2$O$_3$, it was decided to carry out the main calculation of the solid phase for the most significant (and heaviest) part of it (Fe$_2$O$_3$).

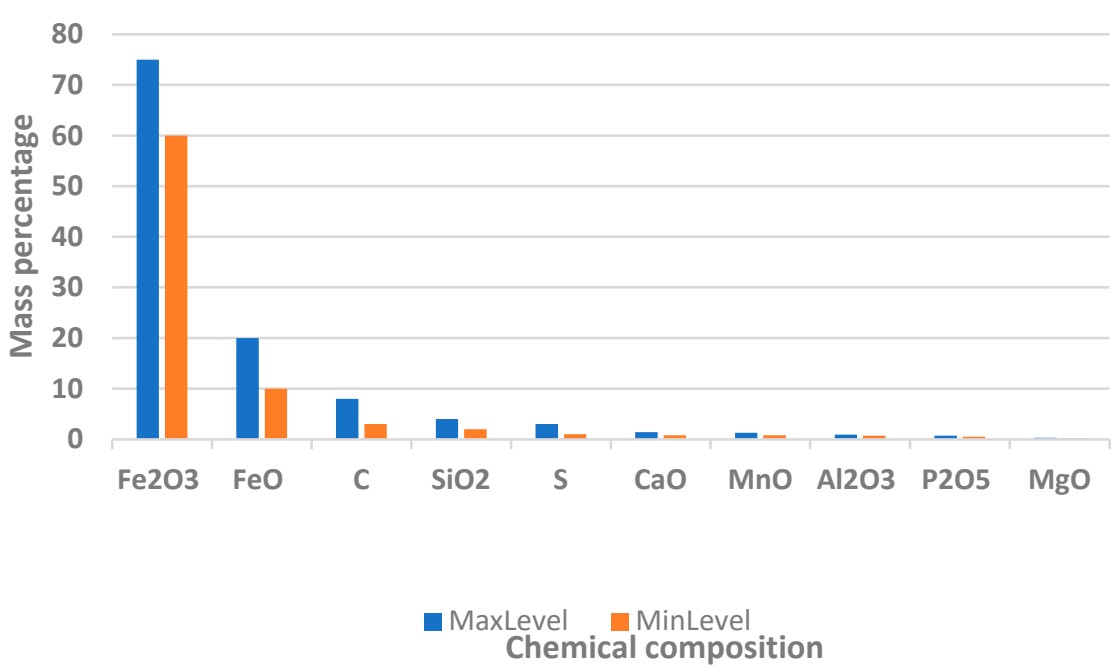

**Figure 4.** The distribution of the chemical composition of dust by weight in percent.

7.  The particles size distribution of the selected dust component is shown in Figure 5.

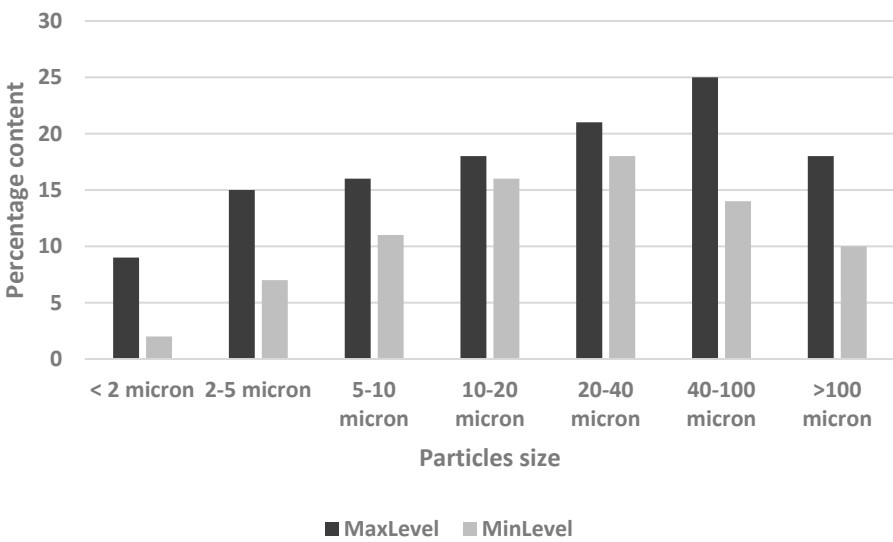

**Figure 5.** Dependence of percentage on $Fe_2O_3$ dust particle size.

As described above, the most unfavorable operating mode was taken for mathematical analysis; tapholes #1 and #4 operate simultaneously, since the gas duct route from tapholes #1 and #4 is the longest and has a greater number of local resistances (bends, transitions, T-joins, etc.) as is shown in Figures 1 and 6.

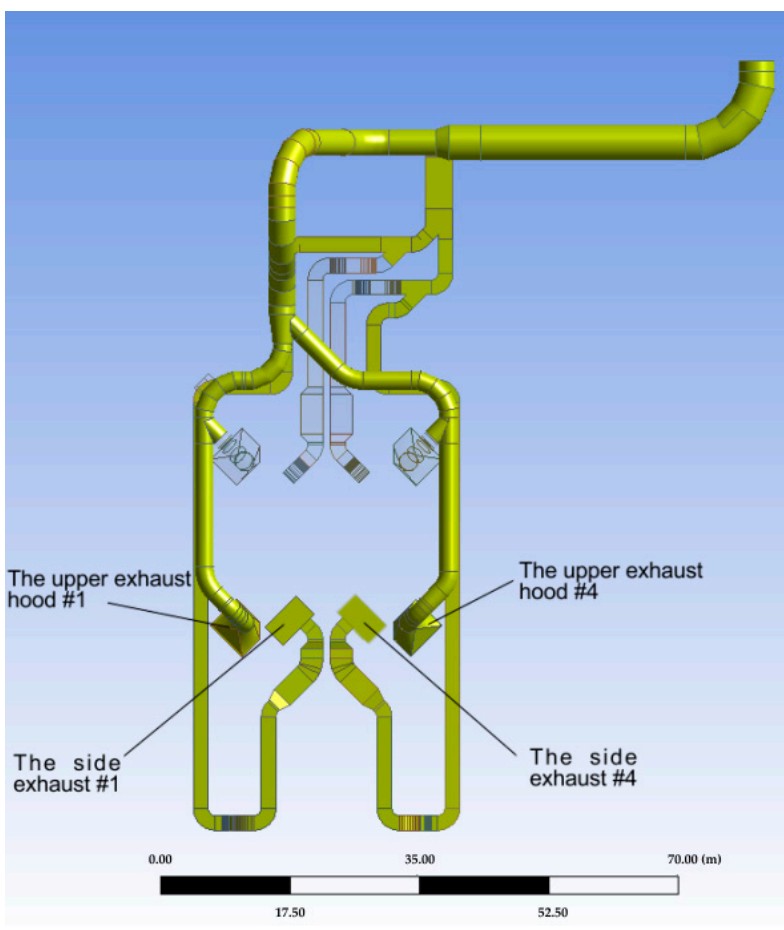

**Figure 6.** Air ducts that are involved in the considered mode of operation.

Through the tapholes under hoods #1 and #4, the flow rate of the dust–gas mixture with a full cross section is set at a velocity of 7 m/s and a value of 280,000 $m^3$/h according to

the "Recommendations for the design of aspiration systems for capturing and cleaning dust and gas emissions from casting yards of blast furnaces with a volume of 2700–5500 m$^3$" [17].

Three-dimensional problem. The most general theoretical description of the flow and heat transfer in the flow path of the system can be performed on the basis of the direct application of the laws of conservation of mass, momentum, and energy to a continuous gaseous medium [18]. The mathematical form of writing the laws of conservation of a viscous gas is the complete system of Navier–Stokes equations, for a three-dimensional region of an arbitrary shape with given boundary conditions with the introduction of simplifying assumptions, can be solved by numerical methods. This approach makes it possible to calculate the characteristics of the system that cannot be determined within the framework of a one-dimensional formulation, in particular, the flow structure, the uneven distribution of the velocity, the maximum temperatures of the structural elements, the distribution of the solid particles and dust of the gas flow, etc.

When performing a three-dimensional aero–gas–dynamic calculation of the aspiration system, the following main assumptions were made:

- The flow of air, blast furnace gas and gas–dust mixture in the flow part of the system is compressible, turbulent, quasi-stationary.
- The difference in specific heat capacities and molecular weights of blast-furnace gas and air is neglected (kg ≈ kv = 1.4; Rg ≈ Rv = 287 J/kg K).

For the simulation of the processes of the flow of a multiphase mixture carrier, in this case, air, a model of gas ducts "with capture" of space in the immediate vicinity of the source of dusting (tapholes #1 and #4) was built. It is shown on Figure 7. This space is limited above and below by platforms. On the other sides, the "walls" of this space were set as transparent—that is, open to the surrounding air.

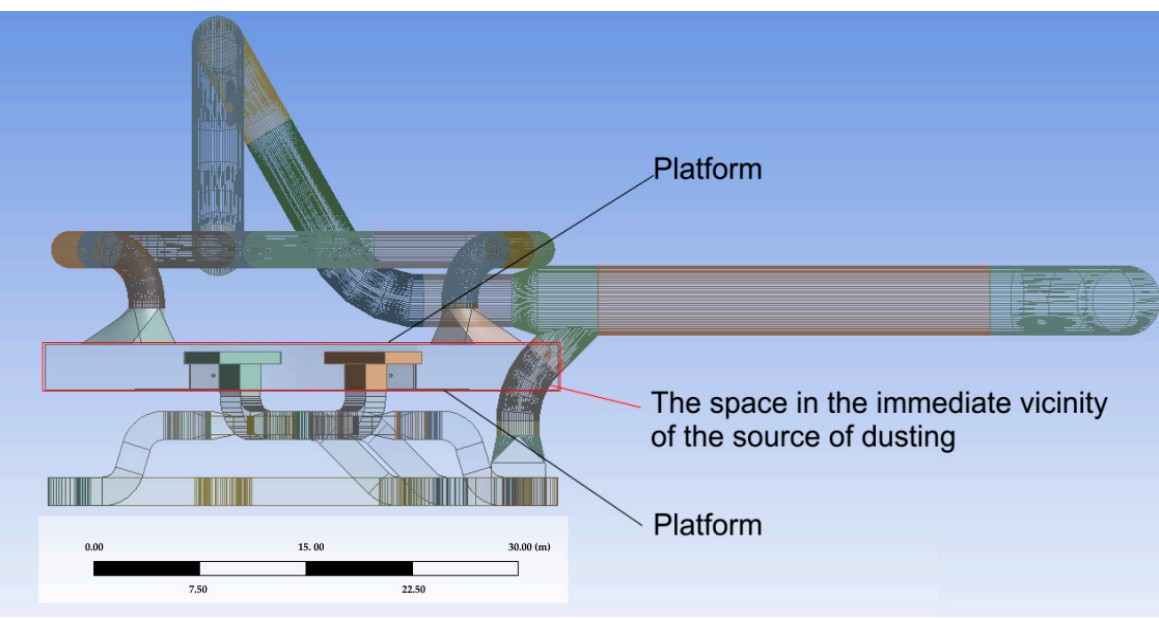

**Figure 7.** Solid 3D model of the flow path of the aspiration system (front view).

For mathematical modeling of a three-dimensional turbulent flow of air, blast furnace gas and the solid phase of the mixture in the flow path of the aspiration system, the ANSYS Fluent computer program was used, which implements the numerical solution of the complete system of Reynolds-averaged Navier–Stokes equations [19–22]. The numerical solution was carried out in the subdomain of space shown in Figure 7, which was covered with a hybrid calculation mesh.

When solving the problem, naturally, the most difficult thing is the simulation of the computational mesh. In the end, the authors came to the following model parameters:

mesh refinement depending on the curvature of the surface, cell volumetric centering tolerance was chosen as fine, and mesh smoothing was chosen as medium. The maximum element size was 0.24 m. Since the aspiration system includes the connection of pipes of various diameters and gas ducts of various rectangular cross sections, it was decided not to set it manually, but to use the software "control" of boundary layers. So, the "automatic inflation" option was used with the following values: the maximum number of layers 5, the Growth Rate 1.2 and, accordingly, mesh refinement for the boundary layers (as shown in Figure 8). With this mesh configuration, the total number of elements was 2,757,970 and the number of nodes (nodes) was 659,305.

The calculations were carried out under steady conditions. The convergence of the solution was controlled using the convergence criterion, which was adopted by a Residual Target = $10^{-6}$. With the described computational mesh, it was necessary to perform about 350 iterations before the convergence of the solution was achieved. The calculation time in this case reached one and a half hours when using a personal computer based on Intel(R) Core(TM) i5-10600KF CPU 4.10 GHz with RAM 16 Gb.

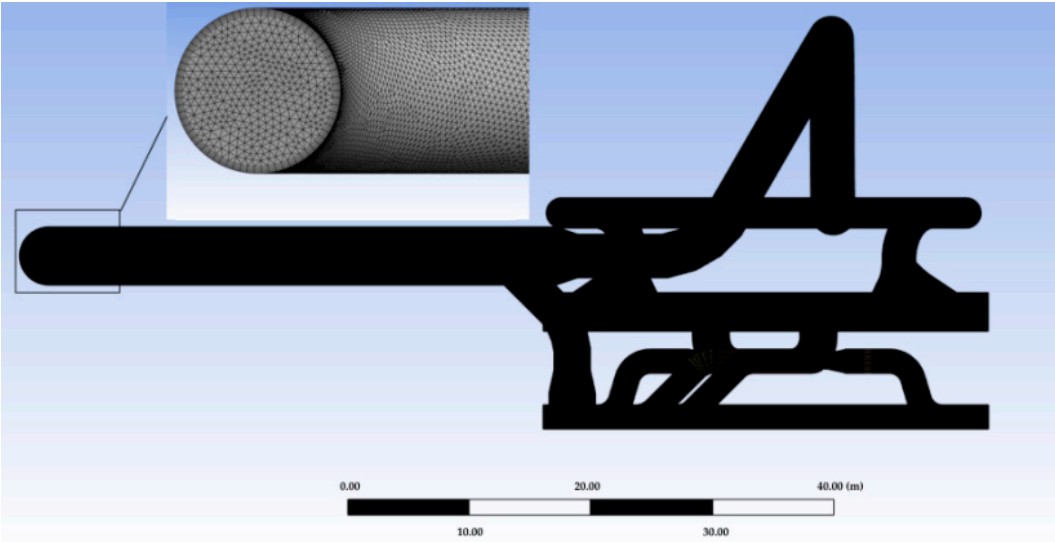

**Figure 8.** The hybrid calculation grid.

Turbulence modeling is carried out using the standard k–$\omega$ turbulence model. Modeling of a multiphase flow is carried out using the Euler equations for multiphase media in a dispersed formulation. The flow simulation is based on the Reynolds-averaged Navier–Stokes equation. The gas phase is modeled according to the ideal gas equations. The mass flow rate of the resulting dust is calculated based on the given dust content of the aspiration flow [23–26].

When modeling the dusty flow, the following restrictions and assumptions were taken:

1.  Near-wall functions were used to take into account the boundary layers not resolved by the grid.
2.  In the areas of the computational domain corresponding to the outlet from the aspiration system, zero excess static pressure, total temperature and turbulence parameters of the gas–air mixture were set.
3.  In the sections of the computational domain corresponding to the inlet of the aspiration system, the total pressure, total temperature, and parameters of air turbulence were set.
4.  The material of the air duct of the suction system was specified as steel, 10 mm thick. Density = 8030 kg/m$^3$, Specific heat Cp = 502.48 J/(kg·K) and Thermal conductivity = 16.27 W/(m·K). Since the solution of the gas–dynamic problem is impossible without taking into account thermodynamic conditions, the wall material is a participant in the heat exchange processes. Since the temperature of the gas–dust

mixture initially reaches a temperature of 350 °C, and the ambient temperature is assumed to be 30 °C, then, in addition to thermal conductivity, there will be heat exchange by radiation from the wall surface, which was also taken into account when solving the problem.

The discretization was performed by the control volume method using the COUPLED algorithm which solves the momentum and pressure-based continuity equations together. Full implicit coupling is achieved through an implicit discretization of pressure gradient terms in the momentum equations, and an implicit discretization of the face mass flux, including the Rhie–Chow pressure dissipation terms.

## 3. Main Results

Figures 9 and 10 show the streamlines of the solid phase in the area of emission from tapholes and at the entrance to the exhaust elements of the aspiration system. Obviously, part of the solid phase, unfortunately, is not captured by the proposed system.

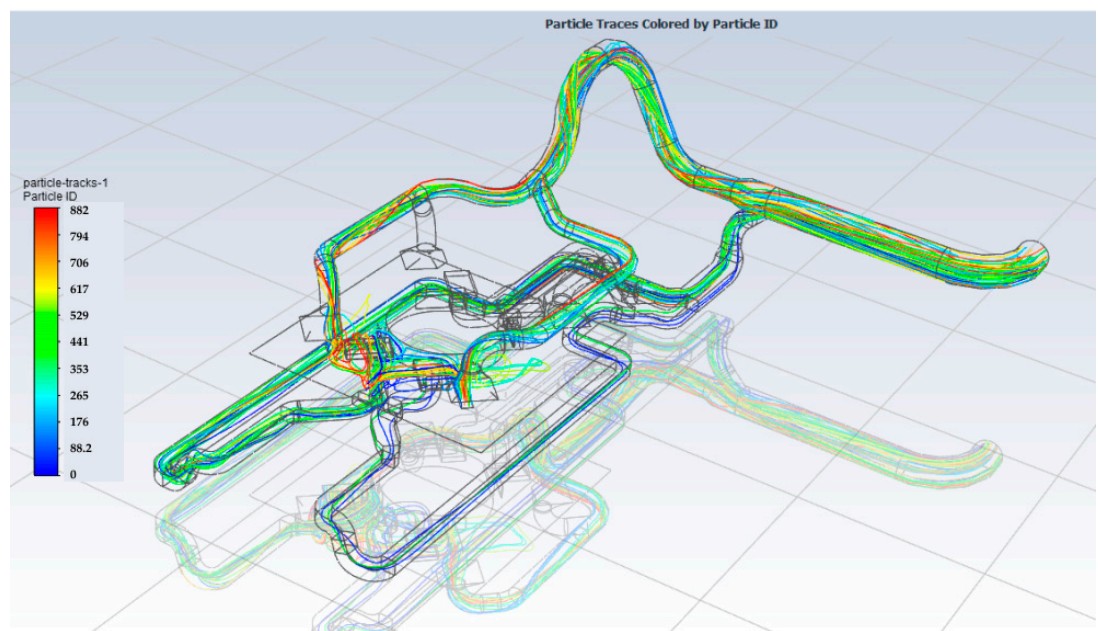

**Figure 9.** Solid phase (dust only) streamlines.

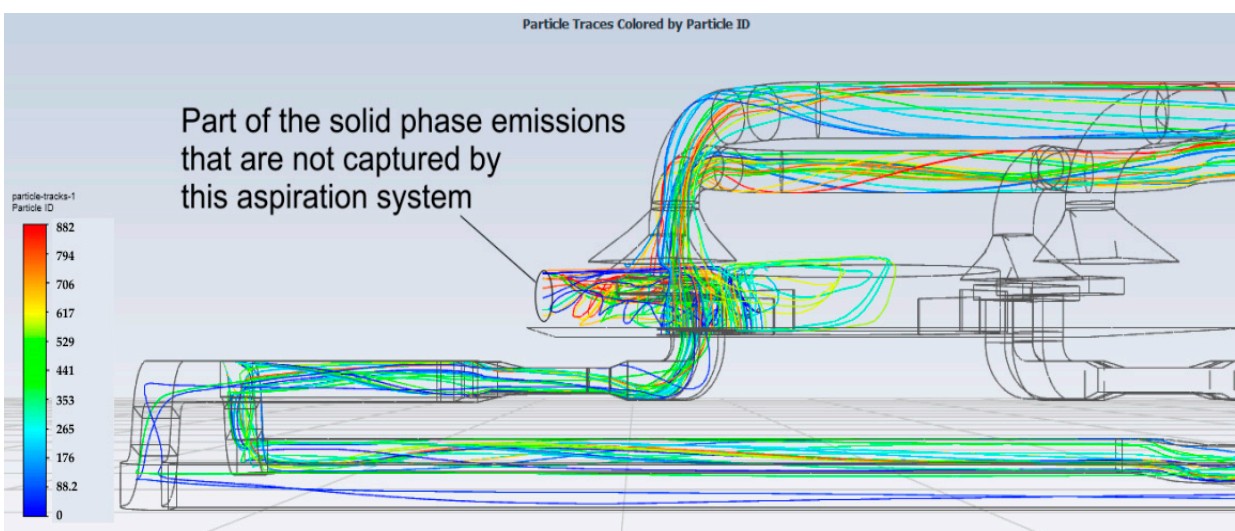

**Figure 10.** Solid phase (dust only) streamlines, focused on the place of "ejection".

The efficiency of the aspiration system is determined on the basis of the integral indicators of the mass flow rates of the solid phase of gas and dust flows coming from sources of pollution and dust flows leaving the aspiration system.

1.  The average integral mass flow rate of the solid phase (dust) at the outlet from pollution sources is 0.233 kg/s + 0.466 kg/s = 0.699 kg/s.
2.  The average integral mass flow rate of the solid phase (dust) at the outlet from the aspiration system pollution is 0.638 kg/s.

The efficiency of dust collection by the aspiration system can thus be estimated as follows:

$$\eta = (0.638/0.699) \cdot 100\% = 91.27\%$$

It can be seen that in the worst case (the most dangerous from the point of view of the operation of the aspiration system for the operating mode in accordance with Figure 6), the efficiency of collecting the solid phase of the gas–dust mixture is more than 90%.

It is clear that the next question for the authors of the work was the following: is it possible to increase the efficiency of dust collection while maintaining the geometry of the proposed aspiration system? Obviously, by varying the productivity of the system, you can significantly affect the efficiency. So, with an increase in productivity of only 10%, the cleaning efficiency of dust collection would be almost 93%, with a further increase— for example, up to 30%, the efficiency will already be at the level of 95%, as shown in Figure 11. However, everything has two faces. It can be seen that with the productivity and efficiency of dust collection of the system, the area-weighted average velocity in the outlet of the aspiration system of the gas–dust mixture also increases and, starting from a 5–10% increase in productivity, this velocity already exceeds 30 m/s and further increases as productivity increases up to 42 m/s. Such a level of velocity of the gas–dust mixture for aspiration systems is considered dangerous from the point of view of the abrasive wear of wall materials [14–16]. Therefore, the authors of the work believe that increasing the efficiency of dust collection by increasing productivity by more than 10% for the proposed geometry is inappropriate and can even be dangerous for materials walls.

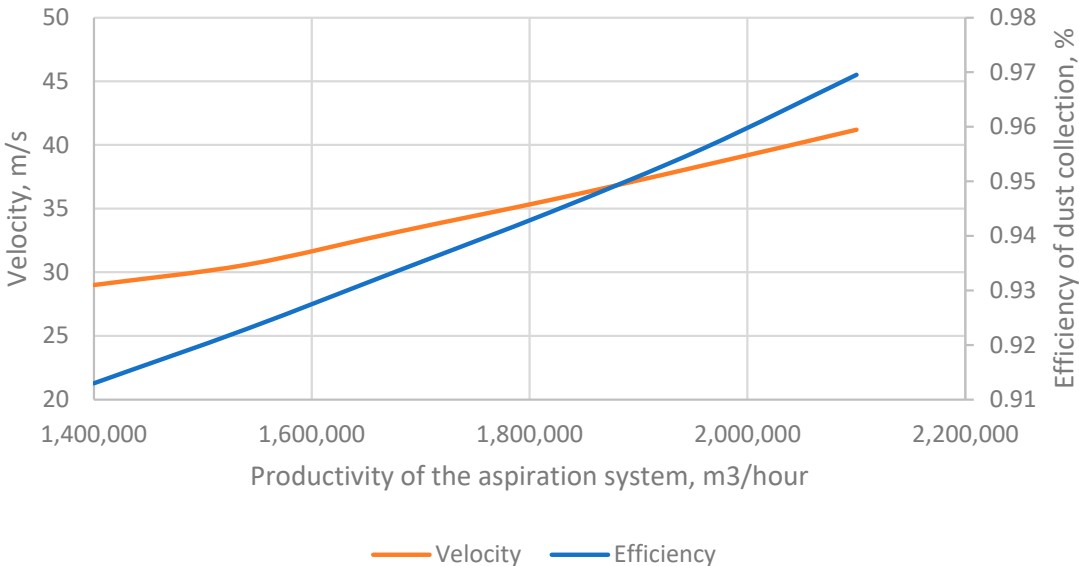

**Figure 11.** Dependence of efficiency of dust collection and velocity of the gas–dust mixture on productivity of the aspiration system.

Figure 12 shows a graphical representation of the volume gradient of the distribution of gas–dust flow velocities through the aspiration system. One-dimensional estimation calculations cannot give a volumetric distribution of velocities over the cross section of the air duct. Thanks to the proposed modeling, it was shown that the velocity of the dust and

gas flow along the length of the aspiration system is in the range of about (20 ... 35) m/s at the inlet to 29.05 m/s at the outlet of the system.

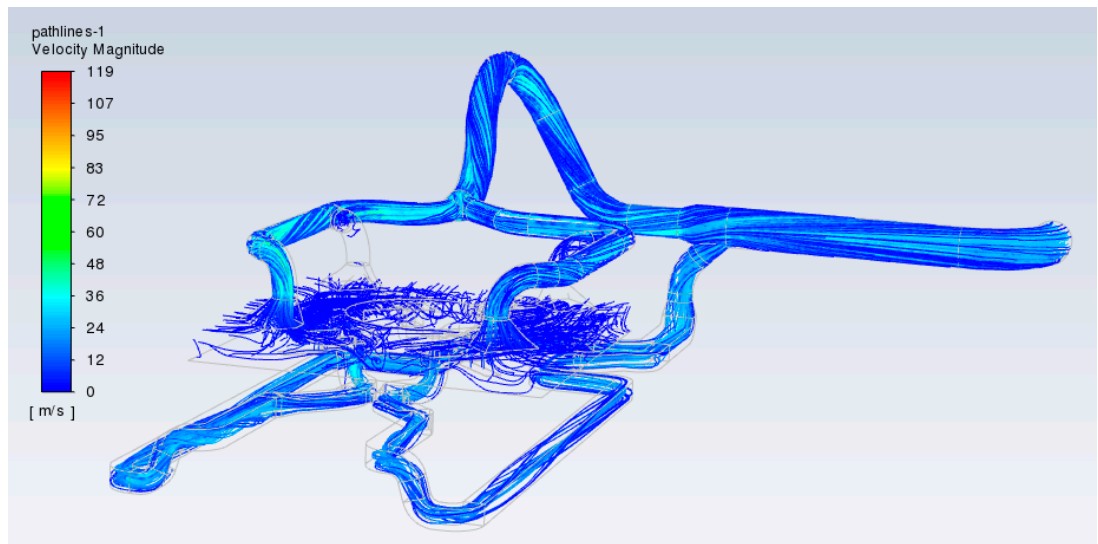

**Figure 12.** The field of velocities streamlines of the gas–dust-air mixture.

Figure 13 shows the average values of velocities in the sections of the gas ducts along the length for all four exhaust lines. Thus, the velocity range excludes the possibility of settling of the solid phase on the surfaces of the air channels of the aspiration system. On the other hand, the velocity of movement of the dust–gas flow is not large enough to cause significant abrasive wear of the structure.

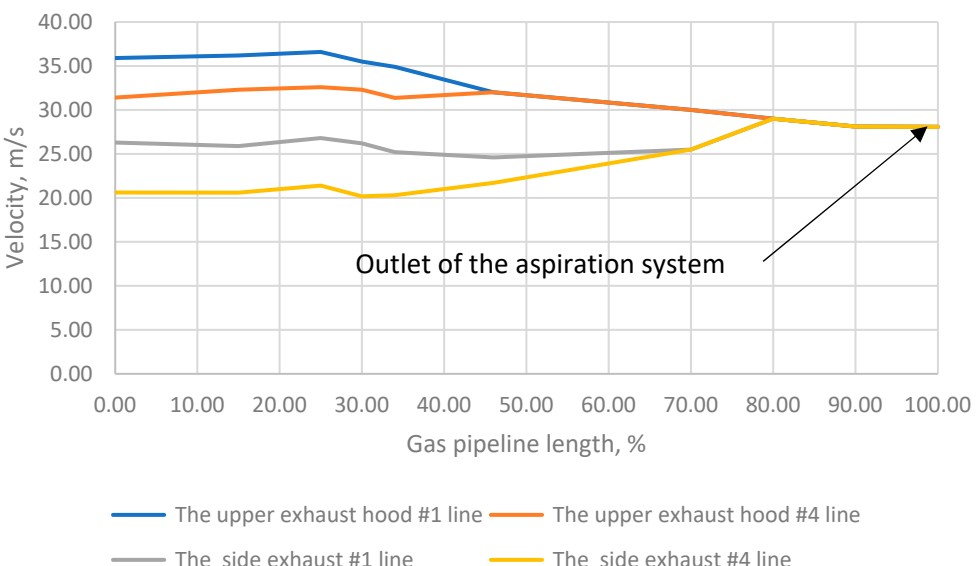

**Figure 13.** The value of the area-weighted average velocity along the length of the gas ducts of the aspiration system.

In the 3D modeling of the design of the aspiration system (walls of gas ducts), an ideal model of a solid body (with steel as the material) was used without taking into account the following: the physical wear of gas ducts during operation, the possible adhesion of dust to the walls and, accordingly, a decrease in the cross-sectional area of the gas duct, which in turn leads to an increase in the resistance of the route and the resistance of stop valves. This means that the actual experimental values of the gas–dynamic flow velocity may be somewhat lower than the calculated ones. Unfortunately, the proposed model

of the aspiration system has not yet been manufactured and, as a result, the calculated values obtained have not been verified, especially with a long operating time, so minor discrepancies can be predicted.

The change in the temperature of the gas–dust mixture in the aspiration system deserves special attention. The fact is that the temperature from the source of the release—the taphole—reaches 350 °C (Figures 14 and 15), and the temperature at the outlet of the system before the proposed simulation could only be estimated (or measured immediately after manufacture and installation). Obviously, the temperature in the space above the entrances and on the platforms is also significant. However, the outlet temperature does not exceed 80 degrees Celsius. This air temperature is acceptable for the gas–dust mixture filtration system, which should be located immediately after the aspiration system. The temperature field of the gas–dust mixture (Figure 16) allows us to additionally select heat-insulating materials or make a decision on heat removal for secondary heat recovery both in industrial and domestic tasks.

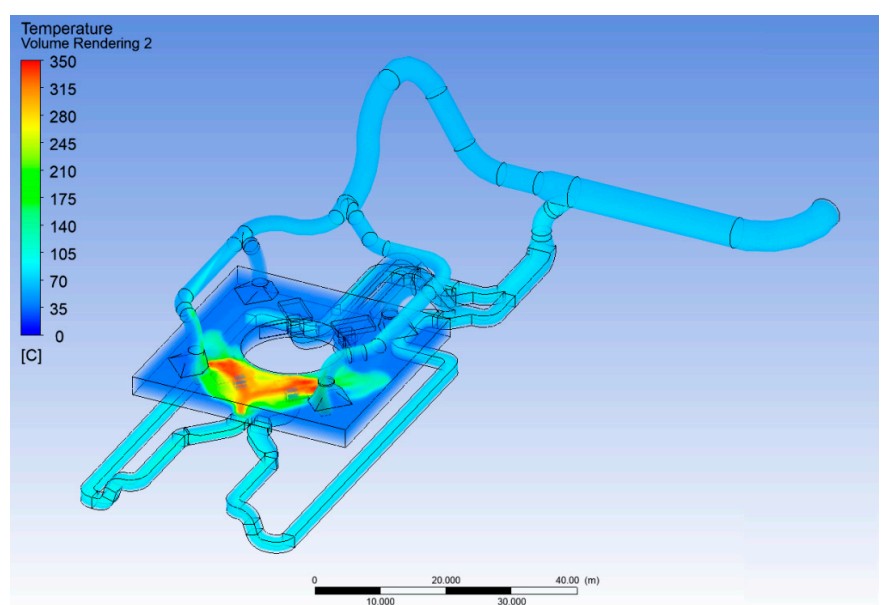

**Figure 14.** Temperature field of the dust–gas mixture.

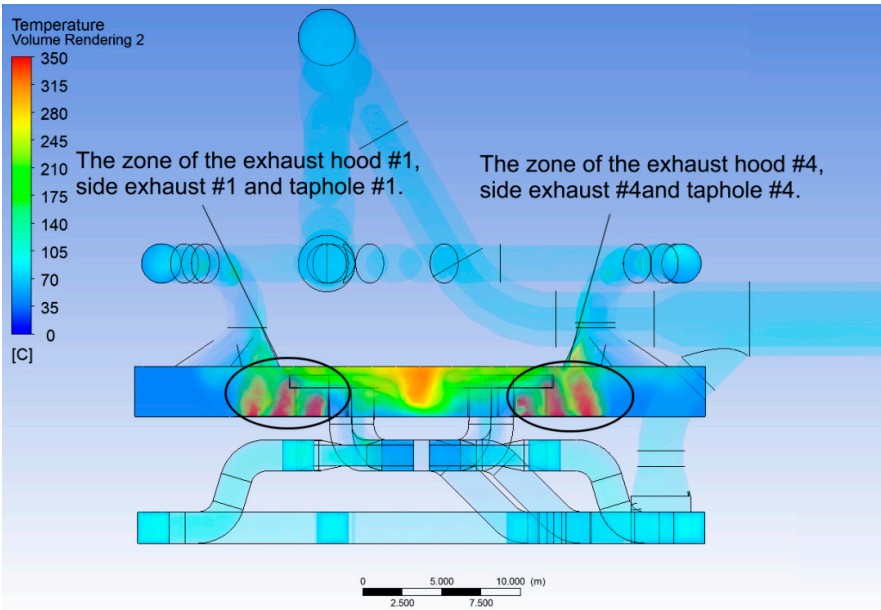

**Figure 15.** Temperature field of the dust–gas mixture, with focus at the "ejection" site.

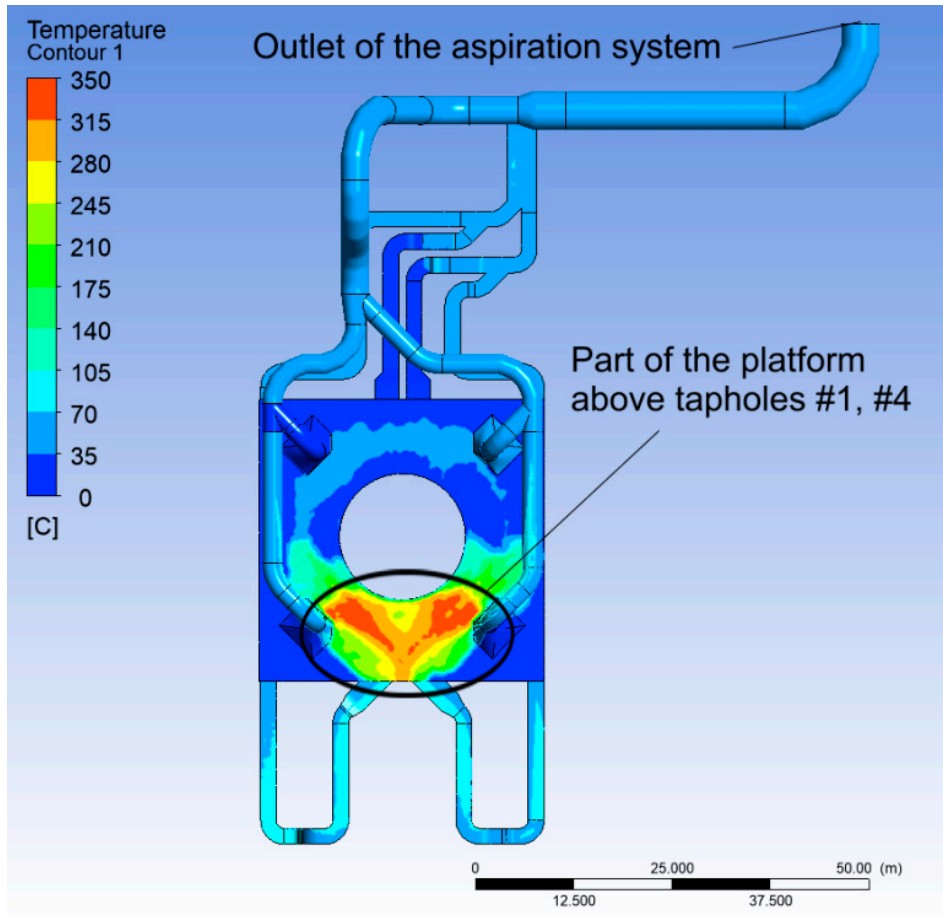

**Figure 16.** Temperature of solid surfaces (walls of the aspiration system).

In accordance with the 3D calculation, the main parameters of the gas and dust flow at the outlet of the aspiration system of the foundry were obtained. They are presented in Table 1.

**Table 1.** The main parameters of the gas–air mixture at the outlet from the aspiration system.

| Parameter | Computed Value |
|---|---|
| Mass flow, kg/s | 361.1 |
| Flow velocity, m/s | 29.5 |
| Flow temperature, 0 °C | 80.5 |
| Mass flow rate of the dust, kg/s | 0.638 |
| Density, kg/m³ | 0.986 |

Unfortunately, as mentioned above, it is not possible to verify the data obtained because, for objective reasons, the construction of the aspiration system at the object of study has not yet been carried out.

## 4. Conclusions

The calculation of the amount of aspiration air necessary for the almost complete removal of the evolved dust–gas–air mixture was carried out.

The designed gas ducts have a precisely calculated cross-sectional area, which makes it possible to keep the velocity of the movement of the dust–gas–air mixture uniform throughout the entire route of the gas duct within the permissible level.

Data on the required aspiration volumes were obtained from the developed design of local suctions.

The study of the dispersed composition and concentration of aspirated dust from the developed design of local suctions was carried out, and the law of distribution of the dispersed composition of aspirated dust was obtained.

The method for calculating the concentration of dust in the air of the working area has been refined when using a complex of dust removal systems, including aspiration systems, in relation to the foundry industry.

It is shown that the efficiency of dust collection of the aspiration system is 91.27%.

It is calculated that the weighted average flow velocity of the gas–dust mixture along the ducts of the aspiration system at the most loaded operating mode is not lower than 20 m/s, which should prevent the gravitational settling of the solid phase on the walls of the gas ducts. On the other hand, this speed does not exceed 35 m/s in a small section of the gas duct, which slightly exceeds the speed range recommended in the literature: from 20 to 29 m/s.

The obtained value of the temperature of the gas and dust flow at the outlet of the aspiration system is 80.5 degrees Celsius, which is below the allowable value for the filter system.

As a recommendation, it can be proposed to change the area of the flow section of the gas duct from exhaust hood #1 to the point of connection of this line with the line of exhaust hood #4. This will reduce the average velocity to more acceptable values. To improve the efficiency of the system, it is recommended to increase the productivity of the proposed aspiration system by no more than 10 percent, which will increase the efficiency to 93.5 percent without significantly increasing the average velocity of the gas–dust flow. However, it is necessary to additionally evaluate the feasibility of increasing productivity, because the cost of increasing productivity can be much higher than the cost of the filter system.

**Author Contributions:** Conceptualization, Y.P. and S.L.; formal analysis, S.L.; investigation, S.L.; methodology, Y.P. and V.K.; project administration, I.B. and V.K.; resources, I.B.; software, S.L.; supervision, S.L. and V.K.; validation, I.B.; visualization, S.L. and I.B.; writing—original draft, S.L.; writing—review and editing, Y.P. and V.K. All authors have read and agreed to the published version of the manuscript.

**Funding:** This research received no external funding.

**Institutional Review Board Statement:** Not applicable.

**Informed Consent Statement:** Not applicable.

**Data Availability Statement:** Not applicable.

**Conflicts of Interest:** The authors declare no conflict of interest.

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
