# Peer review of "Simulation of Multi-Phase Flow to Test the Effectiveness of the Casting Yard Aspiration System"

_computation, doi:10.3390/computation11060121_

Round 1

Reviewer 1 Report

I consider that it is an interesting field of research, nevertheless some issue needs additional information, and some justifications must be added to improve the quality of the manuscript. It does not imply an outstanding innovation in the approach nor in the methodology in this research field, you should focus the attention into the new contributions. There is no validation nor comparison process with experimental data, and it is essential to trust the obtained results. The contributions that this research makes compared to previously existing ones should be highlighted. In general, the quality of the images used in the results must be improved, there are confusing legends, and the lack of definition of the surfaces whose contours are being shown, you don’t define if it is a colored surface, streamlines, etc. The bibliography used is scarce, and not sufficiently updated. Some justifications must be added to improve the quality of the manuscript, in more detail:

Abstract. The abstract information is not enough. The main propose of the research was mentioned, but there are not results nor conclusion details, it must be improved.

Line 56-82: “The proposed modernized aspiration system eliminates these imperfections.” It is not clear for me, did you participate in the process of obtaining the system, that is being evaluated?, or are you verifying the new operation conditions only?. The imperfections of the previous one are mentioned in the introduction, therefore, in any case, the ranges and values provided should be justified and properly referenced.

Figure 5. It is a suggestion, the position of the particle sizes must be in the other sense, growing from left to right, it should be more intuitive. The decreasing order is not natural, and could be confusing.

There is not a mesh convergence study done, the mesh parameters are not defined, and in general I cannot be sure about the mesh quality. You declare 10 million cells, but is it enough? You cannot ensure that the results shown correspond to an adequate convergence. I think there is a broad general consensus on this aspect in the numerical simulation community. The values of cell sizes in the near wall region, first layer thickness, the boundary layer mesh details, the number of layers, etc. are not specified, and this information is a crucial information. The mesh used is not sufficiently well justified, and in general I have serious doubts about the quality of the mesh used in this study.

Figure 7. It doesn't seem like there are 10 million cells, it seems a poor mesh, I don't know where the million cells are, but are not clearly shown. Did you create any refined area? Where? otherwise I don’t understand.

Line 228: “The material of the air duct of the suction system was specified as steel, 10 mm thick” How is this boundary condition affecting? Is there heat transfer activated? Is the thermal conductivity of the metal considered? The conditions and models used are not sufficiently detailed, please correct this aspect.

Line 235-239. This explanation is very confusing for me, it is a mixture of very simple concepts about CFD, with not understandable explanations... please review and modify this paragraph carefully.

Line 271-273. I have serious doubts that this can be affirmed, how is it justified? Based on what? References? Low speeds? not damage to the structure with a dopped particle gas flow at 20m/s? Is there any study on the matter? I have doubts about these sentences.

Simulations were done under steady or transient conditions? It was not commented, please include explicitly this information. The explanation of the methodology of the numerical convergence process is not defined. What was the number of iterations fixed?, or the time of simulation?, did you guaranty residuals below a number?. 

Regarding the computational cost, could you give some information about the used computers? How many nodes are being used? What was the total time for which the simulation was run? You don’t specify, all this information must be indicated.

Figure 11. It is not clear where the velocities shown in the figure are located, in the center section, colored stream lines, this should be specified. This information could be visualized in a cutting plane, or in the central line of the ducts, in any case, you should define where it is shown, it is not clearly understood.

Line 280: “ideal model of a solid body”.  I don't understand what you mean by this definition, please modify, it should be explained better.

Line 284: Where do they get this value from? It should be justified and/or referenced properly.

Author Response

Will please see the attachment.

Reviewer 2 Report

The topic is quite interesting and relevant. In this form the article contains basic design solutions and not enough scientific cover. In this regard, I present the main recommendations for correction:

Introduction. The description of the situation presented is sparse, with most of the citations having been published 10 years ago.

It would be useful to present the latest world solutions in casting area industry. Perhaps the solution presented is already partly in use?

Generally, any industry has problems with pollutant emissions, so it is important to highlight the novelty of the solution idea investigated in this Article. At this point the paper feels more like a re-design rather than a scientific article.

Provide a more detailed methodology for the formation of the computational grid.

Page 8, line 251, point 1, most likely the output should be changed to input. Line 261, why 0.630 instead of 0.638?

No verification phase of the results is presented.

Conclusions. Provide more numerical values of the results.

The final paragraph could consist of a main recommendation based on the results.

Round 2

Reviewer 1 Report

First of all, I would like to acknowledge the effort made by the authors in the detailed and extensive answers, and the effort to include some modifications and clarifications in the document. I consider that the quality of the paper has been improved, and now it could be published.